# Exploring the Shifting Geographical Pattern of the Global Command-and-Control Function of Cities

**Piotr Raźniak** [1,*] , **György Csomós** [2] , **Sławomir Dorocki** [1] and **Anna Winiarczyk-Raźniak** [1]

1   Institute of Geography, Pedagogical University of Krakow, 30-084 Krakow, Poland;
    slawomir.dorocki@up.krakow.pl (S.D.); anna.winiarczyk-razniak@up.krakow.pl (A.W.-R.)
2   Department of Civil Engineering, University of Debrecen, 4028 Debrecen, Hungary; csomos@eng.unideb.hu
*   Correspondence: piotr.razniak@up.krakow.pl

**Abstract:** In recent years, some cities have experienced significant growth in terms of command and control functions of cities, and thus have managed to relocate themselves to a much upscale position in the global economy. The main goal of this study is to examine the command-and-control function of cities and the impact of the relocation of corporate headquarters on a city's command-and-control function. The study examines the changes in the revenues of companies located in selected cities and countries and measure the command-and-control function ("C&C") of cities that well illustrates the strength of cities and countries in the global economy. To achieve our goals, we employ a composite indicator, the Command and Control Index that integrates such fundamental financial data of companies as revenues, profits, market value, and assets. In the analysis, we consider the companies that are listed by Forbes Global 2000. Our findings reinforce that the command-and-control function of the traditional centers of corporate headquarters has been lessening for a while, whereas cities located in developing countries and China in the first place have been occupying an increasing position in the global command and control. Now, we are experiencing the robust growth of Beijing's command-and-control function index, and the decline of that index of the former leaders (i.e., New York, London, and Tokyo). We can also draw the conclusion that the migration of headquarters does not significantly impact the change of cities' command-and-control function. In addition, when relocating the headquarters, most companies have remained within the same country and some of them have not even left the metropolitan area itself. In recent years, the number of those companies that have relocated the corporate headquarters has increased, and they have experienced increase in their revenues as well. To attract more corporate headquarters, cities has to offer an attractive environment for companies which strategy should be supported by such governmental initiatives as the reduction of corporate taxes for relocated companies.

**Keywords:** corporate headquarters; command and control function; functions of cities; Forbes Global 2000

## 1. Introduction

Cities and their ability to impact the economy in an international sense have been examined since the 1960s [1–6]. According to Sassen [7] global cities play a substantial role in the command and control of the economy and constitute global centers of services. It may be argued that the mobility of capital is the most important aspect of globalization. Sassen [8] argues that globalization has led to a situation where physical distance between cities is slowly becoming less important, both in terms of capital and labor mobility. An international system has emerged based on a globalized economy being characterized by intense capital flows and powerful financial organizations in the focal points. Major cities are now able to draw qualified workers and specialists away from other cities thanks to the presence of large corporations. In this sense corporations are creating international linkages and are turning cities into world centers [9], which further accelerates the process of globalization [10]. Since the early 1990s, international economic linkages between major

cities have been growing due to the establishment of affiliates of corporations outside of the home countries containing the corporate headquarters [11].

According to Hall [12], world cities are home to the largest companies that do business across the world. They are also major centers of finance thanks to the presence of major banks, insurance firms, and other providers of financial services. However, an analysis of intercity connectivities only through the prism of companies from advanced producer services sector [3] does not show the full potential of international linkages. Research based on the location of corporate headquarters does not provide a good basis for a hierarchy of cities on the global scale. This type of analysis of cities' economic impact does not fully account for the impact of cities located in developed countries, thus it is important to also pay attention to regional affiliates of corporations [13]. This alternative approach better reflects strategic decisions on plant location and not just issues associated with corporate headquarters [14]. P. Raźniak, S. Dorocki, A. Winiarczyk-Raźniak [15] investigated the spatial distribution of multinational companies across Central and Eastern Europe and found that the region rather contains the regional offices of MNCs and subsidiaries than corporate headquarters. Finally, while the CEE region is less globalized than Western Europe, which makes it an excellent candidate for study in the area of globalization processes [16,17]. The above context suggests that the European system of cities is increasingly characterized by economic polarization [18]. This has led to a strong dependence of cities in Central and Eastern Europe on their counterparts in Western Europe and outside of it [19,20]. The former is considered a peripheral region of Europe, with the exception of a few cities [21].

Most studies produced in the 1970s and 1980s focused on a limited number of cities and their impact on the world economy, but this approach has changed in the last two decades. Current research on intercity linkages and the command-and-control function of cities considers many cities across the world [22–24]. Now second and even third tier cities with regional importance that may have been omitted in previous studies are also in the focus. At the same time, the actual ownership structure of companies in relation to their geographic distribution has also become an issue for some researchers [25].

S. Krätke [26] claims that the use of a large sample represents a good analytical approach given that a narrow group of corporations or cities may lead to omissions across the Global South where intercity linkages are now only at the formation stage. In addition, a broader analysis of intercity connectivities should involve a group of corporations outside of the APS sector [27]. Recently, world cities have been studied mainly in terms of connectivity [4], although there exists a school of thought that indicates that the command-and-control function is the main characteristic of world cities [28,29]. However, J. Allen [30] criticize this point of view for being an oversimplification, as the command-and-control function is too simple to explain relationships between cities in the era of globalization. World cities possess resources and capabilities that make it possible for them to drive and control the economy. They serve as gateways to entire sectors of the economy and are built on practices of the business elites that wield power and employ both motivation and manipulation techniques to prompt business sector participants to make "desirable" decisions.

At the same time, corporate power extends far outside internal decisions and management. The 20th century may also be described as a century of corporate globalization. This phenomenon is also termed neoliberal globalization based on government policy that enables the evolution of large corporations in all parts of the world. Corporate power infiltrates government power by financing a preferred political party, which often leads to control over poorer countries. The command-and-control function defined in this manner exceeds government power and corporations are able to wield a great deal of political power through their economic clout, the consequences of which remain unknown [31]. The World Trade Organization expects the emergence of international efforts by large corporations designed to protect them from the efforts of national governments seeking to regulate corporate power. This is in effect a declaration of corporate rights that stands in stark contrast to the Declaration of Human Rights produced by the United Nations decades ago [32].

Hence one may treat the command-and-control function as a separate idea originating in the assumptions behind the world city and global city. This idea holds that large corporations produce an effect on economic processes and decisions throughout the world. The largest 2000 corporations on the Forbes Global 2000 list were used to study this form of economic impact. The sample covers the full spectrum of economic sectors. Thus, research on the city ceases to focus solely on the financial sector or advanced producer services sector. The large number of companies studied also enables an analysis of every region of the world [33]. The command-and-control function is important for cities experiencing good economic conditions but may also become important in the event of poor economic performance. This inherent risk is relevant in cases of political instability, war, and recurring economic crises affecting the world economy every few years. In such cases it is important to assess the level of immunity of the command-and-control function to crisis events that will affect a city from time to time [34].

Today there are three main regions characterized by many cities performing the command-and-control function of the world economy: Western Europe, the United States, and East-Asia (with the major role of China, Japan, and South Korea). Emerging regions include the Middle East and Brazil [35]. The significance of the command-and-control function is also growing in Asian cities. For example, the 2008 global economic crisis strongly affected Tokyo with a large decline in economic power, but the effect on Beijing was much smaller [36]. The most important cities in terms of the command-and-control function in 2012 were New York, London, Tokyo, and Paris. However, in 2015 all the above cities were outpaced by Beijing due to a strong acceleration of growth in its financial, energy, and construction sectors [37]. Growth in command-and-control functions values also shifted from Western Europe to Central and Eastern Europe following the 2008 global financial crisis. This was mainly due to outsourcing shifts in the IT and energy sectors after 2006 year [38].

The magnitude of the command-and-control function of cities changes over time. Studies are available that examine this pattern of change [39] along with changes in the number of corporate headquarters and corporate revenue [40]. Other works focus on the command-and-control index as a way of measuring the magnitude of the command-and-control function [22]. It is reasonable to infer that some cities will be able to attract new investment as well as new corporations by offering a package of incentives such as excellent IT and communications, good capital markets, business support institutions, and availability of qualified workers [41].

Given the ongoing process of globalization and the increasing relevance of the command-and-control function in the global economy, it is important to assess whether this new relevance is due to growth at companies already present in a particular city or due to the establishment of new corporate headquarters in the city. For example, IBM moved its corporate headquarters from Hartford (Connecticut) to Boston (Massachusetts) resulting a decline in the C&C function of Hartford and an increase in Boston. In this case Boston is experiencing an increase in the power of its C&C function not only due to the presence of other corporations, but also due to the arrival of a new corporate headquarters. In addition, the economic strength of a city may grow because one of its corporations may acquire a corporation in another city. There are currently no studies examining the impact of the relocation of corporate headquarters on a city's command-and-control function. The present study aims to fill this gap in knowledge by determining the current magnitude of the command-and-control function for cities across the world and by tracking changes in the location of corporate headquarters for entities listed by Forbes Global 2000 and their impacts on the C&C function of cities.

There are many studies available on cities, but there are no studies analyzing the C&C function for countries and regions with comparable economic potential. Thus, the present study provides an analysis for the United States, China, Japan, and the European Union (EU).

## 2. Data and Methods

The study uses financial data for the largest corporations from the Forbes Global 2000 list [42]. The data includes revenues, profits, assets, and market value. Revenue is a company's income before expenses, and profit is income after expenses. Forbes compiles the annual Forbes 2000 dataset by using the same methodology and according to same standards. Naturally, it would be highly difficult to compare the raw data contained in the 2006 and 2018 datasets because in this short period of time the cumulative inflation of the U.S. dollar was equal to roughly 25%. Therefore, we had to introduce an index that would help compare cities' command-and-control function without relying on values in U.S. dollars. Company data were downloaded from annual Forbes 2000 datasets and then investigated in terms of the geographic location of the companies' headquarters. The firms found on this list generate the command-and-control function of cities across the world [22]. Corporate financial data were assigned to the city home to corporation's headquarters. A total of 2000 firms were assigned to several hundred cities each year. The number of cities varied from 353 in 2006 to 408 in 2012.

Corporate financial data were assigned to the city home to a corporation's headquarters. A total of 2000 firms were assigned to several hundred cities for each year. The number of cities varied from 353 in 2006 to 408 in 2012. Data were added for cities found within the same metropolitan area. Given the dominance of the central city in each given metropolitan area, all companies present within its boundaries were treated as belonging to the central city. The boundaries of major metropolitan areas were identified using criteria employed in each country studied. For example, in the United States, boundaries established by the U.S. Census Bureau [43] were used in the study.

The power of the command-and-control function is expressed herein using the command-and-control index (CCI), which is calculated for every city in the following manner (Csomós 2013):

$$\mathrm{CCI}_{x,y} = \sum_{i=1}^{n_{x,y}} \frac{\mathrm{R}_{i,x,y} + \mathrm{A}_{i,x,y} + \mathrm{P}_{i,x,y} + \mathrm{MV}_{i,x,y}}{4} \tag{1}$$

- $\mathrm{R}_{i,x,y}$ = proportion of revenues
- $\mathrm{A}_{i,x,y}$ = proportion of assets
- $\mathrm{P}_{i,x,y}$ = proportion of profits
- $\mathrm{MV}_{i,x,y}$ = proportion of market value
- $i$ = number of companies headquartered in a city in a given year ($i = 1, \dots, n_{x,y}$);
- $n$ = total number of companies headquartered in city $x$ in year $y$

The command-and-control index (CCI) was calculated as the average of the percentage share of four financial indicators (revenues $\mathrm{R}_i$, assets $A_i$, profits $P_i$, market value $\mathrm{MV}_i$) for all the examined cities. The total value of CCI for a given year for all the studied cities is 100.

Revenues consist of funds acquired by a company from the sale of goods and services. Many accountants use the term "sales" and "revenues" interchangeably. Revenue does not have to mean an influx of cash. A part of revenue is paid in cash, while another part may be paid using credit. The value of goods and services sold accurately reflects the level of economic activity at a given company. The economic strength of a city may also be computed using corporate revenues data, as in the case with Taylor and Csomós [40]. Our study examines the economic strength of leading cities and entire countries as well. However, it should be stated that the dynamics of revenues does not take into account inflation, which may differ from country to country.

Basically, there are two publicly available datasets that contain data on companies: Fortune 500 and Forbes 2000. As their names suggest, Forbes 2000 ranks four times more entities than Fortune 500. It is true, however, that the financial data they list roughly overlaps. Naturally, Forbes 2000 (also Fortune 500) is characterized by a critical limitation: it contains information on publicly listed companies only. This also means companies whose stocks are not traded on stock exchanges are not considered by Forbes and are not included in Forbes 2000. For example, the list does not contain data on Lego, the world's

largest toy company, Robert Bosch AG and Koch Industries, two of the world's largest industrial conglomerates, and IKEA, the world's largest furniture retailer. This is a major weakness of Forbes 2000 as well as Fortune 500, and a weakness of every single piece of research that focuses on measuring cities' command-and-control function. Unfortunately, there are no other options than the use of datasets on publicly listed companies because it is not possible to procure data on privately-held companies.

## 3. Changes in the Command-and-Control Function in the US, Japan, China, and the EU

In the period of 2006–2011, New York was the leading city in terms of the command-and-control function, except for 2009 when Tokyo surpassed it. The first four cities in this ranking at the time were New York, Tokyo, London, and Paris. They would change places, but overall would remain the top four cities in the world. In 2006, the number five city was Dallas, but this was the only year when this city would rank this high.

Beijing was ranked fifth in 2007 and remained ranked fifth until 2011. Beijing has continued to climb the ladder since 2011. In 2012, it was ranked number four, in 2013–2014 number three, and in 2015–2018 it became the highest ranked city in terms of command-and-control. In 2012–2016 New York was ranked second, and in 2017–2018 third (behind Tokyo). In 2013–2018 London was ranked number four and Paris as number five. Thus, one may observe a dynamic rise in the significance of Beijing along with a corresponding decline for New York and stabilization for London and Paris (Table 1).

**Table 1.** Top five cities using the command-and-control index in the years 2006–2018.

| 2006 | | 2007 | | 2008 | | 2009 | | 2010 | |
|---|---|---|---|---|---|---|---|---|---|
| City | CCI | City | CCI | city | CCI | City | CCI | City | CCI |
| New York | 8.90 | New York | 8.89 | New York | 7.48 | Tokyo | 7.80 | New York | 7.28 |
| Tokyo | 8.09 | Tokyo | 7.34 | London | 6.65 | New York | 6.57 | Tokyo | 6.75 |
| London | 6.55 | London | 6.94 | Tokyo | 6.54 | Paris | 6.33 | London | 6.34 |
| Paris | 5.69 | Paris | 5.86 | Paris | 6.32 | London | 5.92 | Paris | 6.23 |
| Dallas | 2.20 | Beijing | 2.22 | Beijing | 3.04 | Beijing | 3.68 | Beijing | 4.26 |
| 2011 | | 2012 | | 2013 | | 2014 | | | |
| city | CCI | city | CCI | city | CCI | city | CCI | | |
| New York | 6.80 | Tokyo | 6.95 | Tokyo | 6.92 | Tokyo | 6.86 | | |
| Tokyo | 6.37 | New York | 6.52 | New York | 6.49 | New York | 6.60 | | |
| Paris | 5.72 | London | 5.73 | Beijing | 6.41 | Beijing | 6.52 | | |
| London | 5.66 | Beijing | 5.43 | London | 5.51 | London | 5.30 | | |
| Beijing | 4.98 | Paris | 5.03 | Paris | 4.89 | Paris | 4.72 | | |
| 2015 | | 2016 | | 2017 | | 2018 | | | |
| city | CCI | city | CCI | city | CCI | city | CCI | | |
| Beijing | 7.61 | Beijing | 7.83 | Beijing | 7.98 | Beijing | 8.07 | | |
| New York | 6.67 | New York | 6.87 | Tokyo | 7.12 | Tokyo | 7.13 | | |
| Tokyo | 6.28 | Tokyo | 6.81 | New York | 6.64 | New York | 5.87 | | |
| London | 5.14 | London | 4.77 | London | 4.28 | London | 4.67 | | |
| Paris | 4.50 | Paris | 4.15 | Paris | 4.04 | Paris | 4.08 | | |

In 2006, corporations headquartered in the top five cities out of 350 to 400 cities examined every year the world generated 7200 billions USD in revenues, while in 2018 this was 11,200 billions USD. This is an increase of 54.6% in 12 years, this despite the occurrence of the 2008 global financial crisis. The highest corporate revenues in the period 2006–2018 were noted for Tokyo, with revenues since 2009 exceeding 3000 billions USD. On the other hand, corporations headquartered in Paris, New York, and London generated between 1500 and 2000 billions USD in revenue in the years 2006–2018, and no major changes in

revenue were noted. The trend for Beijing was quite different, with a major increase in revenues from 207 billions USD in 2006 to 2800 billions USD in 2018—for an increase of +1.353%. The 2008 global financial crisis impacted most negatively companies operating out of New York and London, where a decline in revenues could still be observed 3 to 4 years after the start of the crisis. On the other hand, the crisis was nowhere to be seen in Beijing, revenues growing every year, and it was Beijing that contributed most to the growth in the top 5 cities (Figure 1). The Forbes global 2000 list doesn't contain the same number of companies for each city in each year. In 2006, the Forbes list included 96 of New York-based companies and 86 in 2016 year [44]. The output of New York decreased in that time, but it is not just because of the lessening performance of the companies but also because the number of New York based companies.

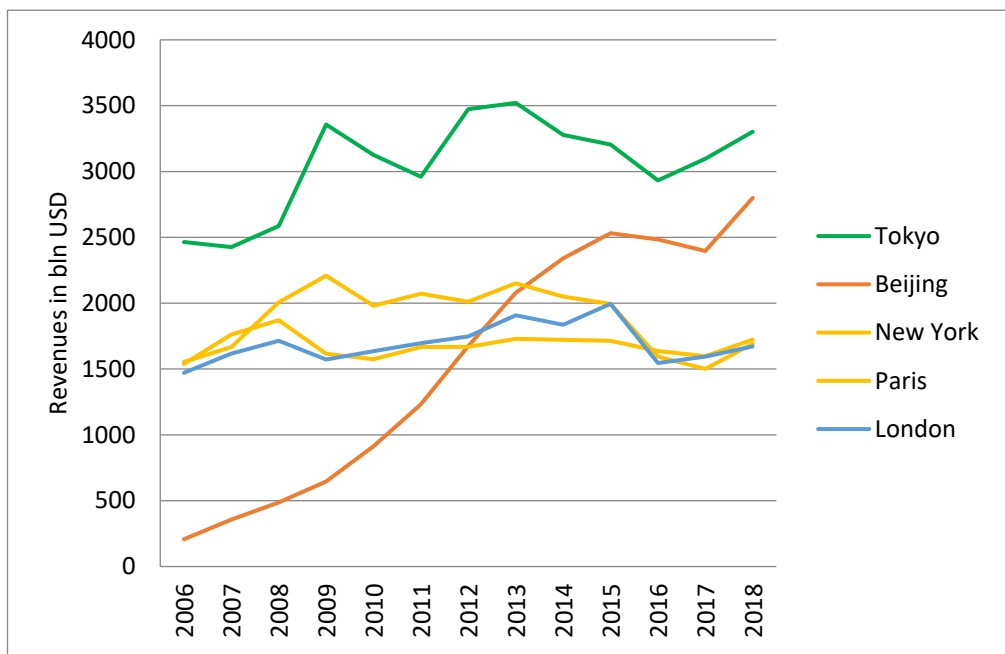

**Figure 1.** Corporate revenues for the top 5 cities in the world in terms of the command-and-control function in the years 2006–2018 (Top 5 cities based on the classification from 2018).

An analysis of revenues for selected countries and the EU leads to two pairs following similar trends. A roughly 20% increase in revenues occurred in Japan and the EU in the period 2006–2013, followed by a small decline since 2017. The trend was more favorable in Japan, where the entire studied period was characterized by growth at 30.54%, while in the EU it was only 15.29%. Despite the lower growth rate in 2018 in the EU, revenues stood at 8938 billions USD, while in Japan at 4460 billions USD. Another pair to be compared was the United States and China. The United States experienced a general growth trend, except for 2009–2010, most likely due to the global financial crisis which in fact began in the USA. The crisis did not affect revenues in China to any observable extent. Revenues grew rapidly in China throughout the studied period, with a small slowdown in the years 2015–2017. This translates into an increase in revenues for companies with headquarters in China from 391 billions USD in 2006 to 5728 billions USD in 2018. American companies still generated a much higher total Revenue (from 9163 billions USD in 2006 to 12,441 billions USD in 2018), but the rate of growth of Chinese companies (1363%) by far exceeded that of US-based firms (35.77%). This created a situation where the country dominant in 2006—the United States, with revenues at 44.21% of the total revenues of the studied group—declined by 2018 to 39.41%. Analogically, the share of China's revenues increased tenfold from 1.89% in 2006 to 18.15% in 2018—reaching 46% of the revenues of the United States in that year (Figure 2).

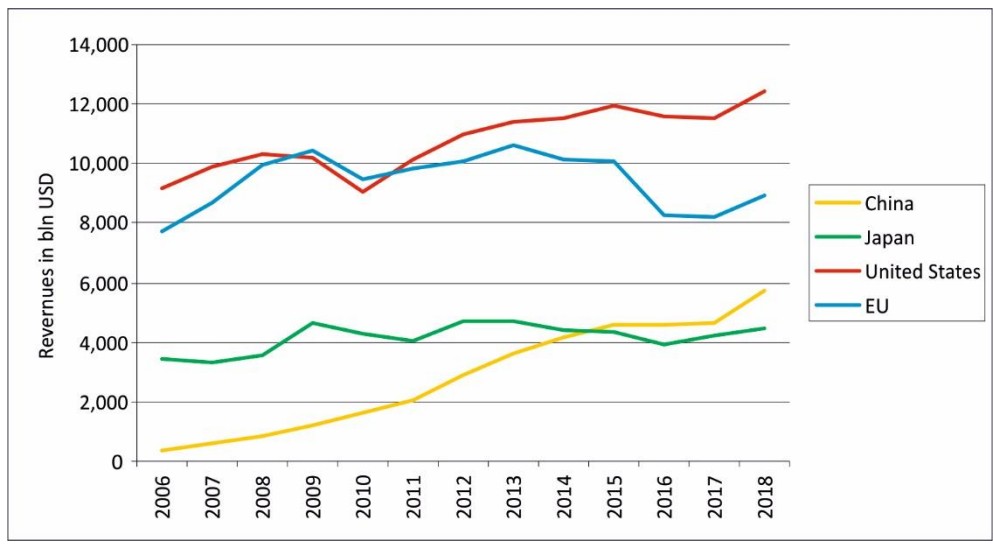

**Figure 2.** Total revenues in 2006–2018 in billions USD.

An analysis of the geographic location of all 2000 Forbes companies and their effects on the command-and-control function of cities, one concludes that American cities remain dominant, yet with a decline in their overall significance from 37.95 to 29.68 CCI units (decline of 21.8%) in the years 2006–2009. An increase was noted for the USA in the years that followed. The United States has been again leading Japan, China, and the EU in recent years. The command-and-control function of Japanese cities declined marginally and remains below 10.0 on the CCI scale during the study period since the 2008 global financial crisis. The trend is different for the EU characterized by the largest decline in the studied period at 26.01%. In the years 2016–2018 one may observe a reversal in the EU's negative trend and a reversal of the USA's positive trend, which has led to a smaller index difference between the two regions. A completely different trend was observed for China. Despite a host of global economic problems, China's CCI increased from 1.58 in 2006 to 16.69 in 2018 for a total growth rate of 1.056%. China outpaced Japan in 2013 (Figure 3).

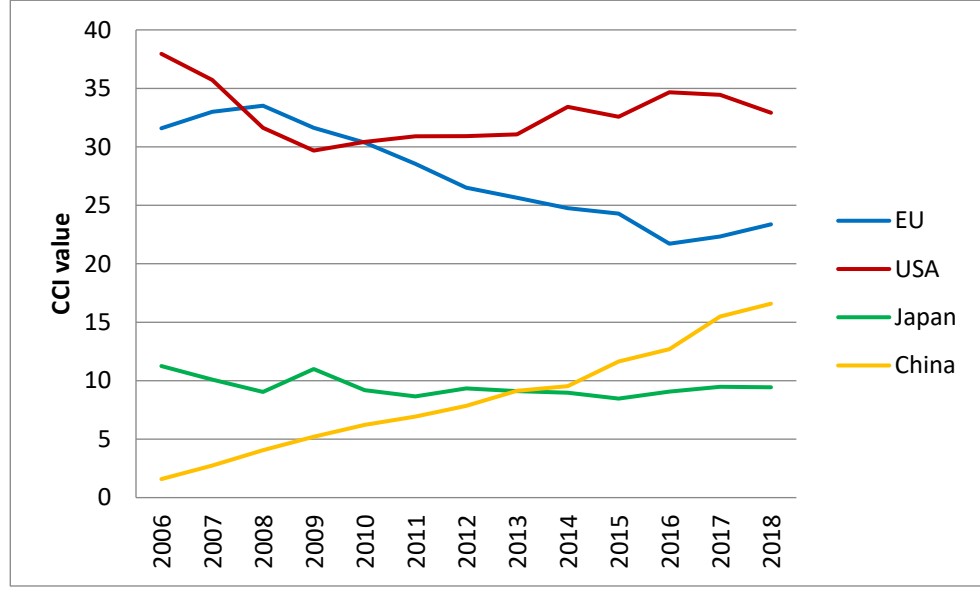

**Figure 3.** Command-and-control index values for selected countries and regions.

## 4. Changes in Corporate Headquarters Location and Their Impact on Revenues Gained by Firms Operating in Selected Cities and Their Command-and-Control Function

Only 53 companies changed their headquarters location in the period 2006–2018. Their total revenue was small at only 0.16% of all revenues of all the studied firms in the study period. Not one company changed its headquarters location in 2006 and 2008. In 2010 revenues of those companies stood at 92.7 billions USD, only to strongly decline in the following year. A growth trend can be observed from 2011 to 2018. Annual revenues exceeded 100 billions USD starting in 2016 (Figure 4).

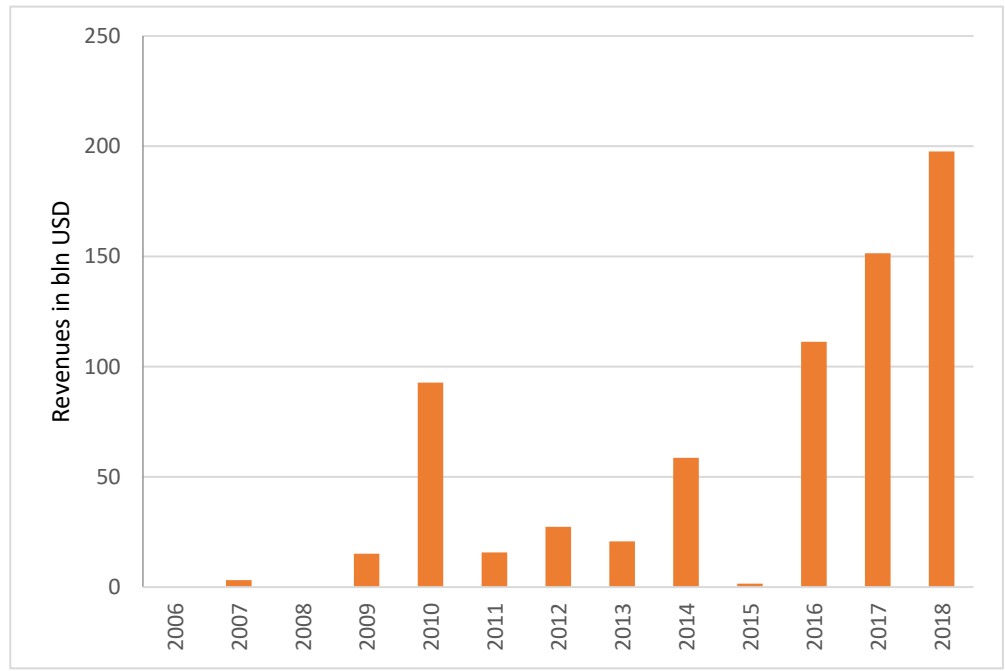

**Figure 4.** Revenues of firms that changed their headquarters location.

Given the rather small number of firms that did change their headquarters location (Figure 4), the study examines changes in two time periods: 2006–2010 when revenues varied significantly, and the years 2011–2018 characterized by a general growth trend. Total revenues of these firms in the first time period stood at 126.71 billions USD or 25.31 billions USD per year, on average (Table 2).

**Table 2.** Total revenues of firms that changed their headquarters location in 2006–2010 in billions of US dollars.

| Countries And Regions Migrated from | Countries and Regions Migrated to | | | | |
|---|---|---|---|---|---|
| | Bermuda | Cayman Islands | Russia | United States | Total |
| Switzerland | 44.48 | 19.01 | 0 | 0 | 63.49 |
| EU | 46.51 | 11.1 | 3.16 | 2.45 | 63.22 |
| Total | 90.99 | 30.11 | 3.16 | 2.45 | 126.71 |

The number of firms changing their headquarters location in the years 2011–2018 was larger (Table 3). Their revenues stood at 568.56 billions USD or 71.07 billions USD per year, on average, for an increase of 181% in relation to the years 2006–2010. Most changes in headquarters location in the years 2011–2018 occurred within the same country or UE. Firms in China, Japan, Taiwan, and the United States changed their headquarters locations only within their own borders. The situation was somewhat different in the EU, where companies with total revenues at 127.79 billions USD had relocated there from the

USA. Despite this fact, firms generating 63.2% of all revenues relocated only within EU member states.

**Table 3.** Total revenues of firms that changed their headquarters location in 2011–2018 in billions of US dollars.

| Countries and Regions Migrated from | Countries and Regions Migrated to | | | | | | | | | | |
|---|---|---|---|---|---|---|---|---|---|---|---|
| | Bermuda | Channel Islands | China | Japan | Papua New Guinea | Russia | Switzerland | Taiwan | EU | USA | Total |
| Australia | 0 | 0 | 0 | 0 | 1.6 | 0 | 0 | 0 | 0 | 0 | 1.6 |
| Bermuda | 1.8 | 0 | 0 | 0 | 0 | 0 | 0 | 0 | 10.4 | 0 | 12.2 |
| China | 0 | 0 | 20.89 | 0 | 0 | 0 | 0 | 0 | 0 | 0 | 20.89 |
| Japan | 0 | 0 | 0 | 12.64 | 0 | 0 | 0 | 0 | 0 | 0 | 12.64 |
| South Africa | 0 | 0 | 0 | 0 | 0 | 0 | 0 | 0 | 4.8 | 0 | 4.8 |
| Switzerland | 0 | 0 | 0 | 0 | 0 | 0 | 0 | 0 | 21.5 | 23.62 | 45.12 |
| Taiwan | 0 | 0 | 0 | 0 | 0 | 0 | 0 | 19.8 | 0 | 0 | 19.8 |
| EU | 0 | 6.92 | 4.6 | 0 | 0 | 15.8 | 6.4 | 0 | 276.9 | 127.79 | 438.41 |
| USA | 0 | 0 | 0 | 0 | 0 | 0 | 0 | 0 | 0 | 13.1 | 13.1 |
| Total | 1.8 | 6.92 | 25.49 | 12.64 | 1.6 | 15.8 | 6.4 | 19.8 | 313.6 | 164.51 | 568.56 |

Firms relocated their headquarters mostly from the USA and tax havens such as Bermuda and the Cayman Islands. Companies' exit from the United States is caused mainly by the search for cheap labor and for highly qualified specialists. A good example of this is the biotechnology industry, which has been migrating to Switzerland and Ireland in recent years [45]. A certain glaring example of this is the case of President Obama persuading Pfizer from leaving the United States in the year 2016. The company had wanted to move to Ireland to merge with the Allergan company, which has its headquarters there. Pfizer's intention was motivated by the decrease in corporate taxes. One sector that does relocate its corporate headquarters relatively often is the high-tech sector. This is often due to non-economic issues including the use of global production networks [46]. The opposite is true of the automobile industry, which tends to relocate factories and not headquarters [47,48]. Another example is light industry [49]. Yet another process is the abandonment of tax havens. This is mainly due to difficulties arising from the actions taken by many governments to reduce the influence of tax havens, which has become even more important since the 2008 global financial crisis [50]. Both the trend towards government restrictions on tax havens and the trend to establish specialized lower tax zones in Europe has led to an increase in the number of company relocations to Europe.

Gigantic growth rates may be observed in China both in terms of revenues and CCI values due to the expansion of domestic companies. Tables 2 and 3 show the negligible impact of company headquarters migration to China on corporate growth in this country.

A total of 13 companies moved their headquarters to another city in the years 2006–2010. The greatest beneficiary was Dublin (Ireland), with four companies migrating their HQs there—three from Hamilton in the Bermudas and one from George Town in the Cayman Islands. Their total revenue stood at 51.61 billions USD. Another four companies moved to Zurich (Switzerland)—two from Hamilton and two from George Town. Their total revenue stood at 33.84 billions USD. Dublin and Zurich were the clear winners in terms of the number of in-migrating companies (61 and 54, respectively) as well as in terms of their share of revenue (72.17%). Switzerland was an attractive destination in the studied period—another two companies migrated from Hamilton and George Town to Luzern, and one from Hamilton to Lausanne. One of the key cities of Europe—London—was only able to attract one firm, with a mere 2.45 billions USD in revenue. On the other hand, leading tax havens such as Hamilton and George Town were biggest losers (−7 and −4, respectively). It is also noteworthy that corporate HQ migrations occurred only between cities in the USA, the Bermudas, Cayman Islands, and Europe (Figure 5).

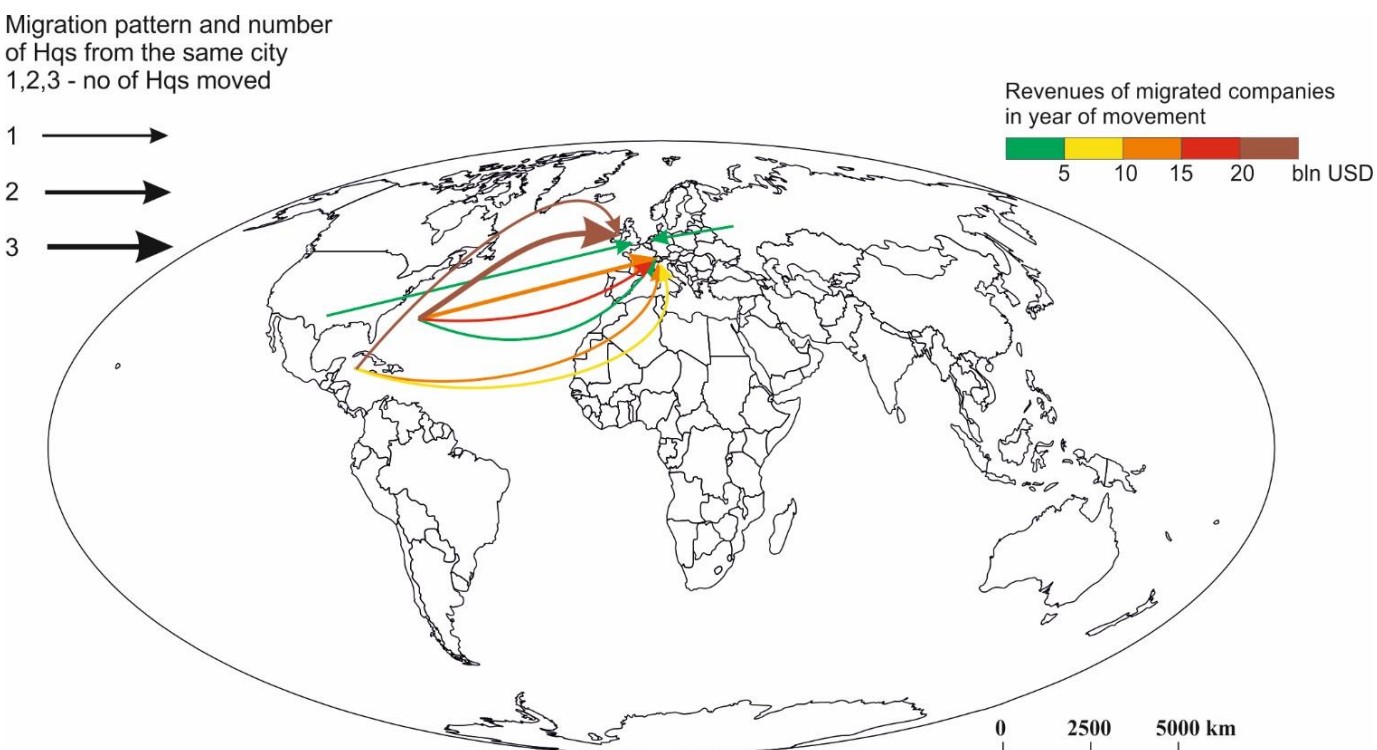

**Figure 5.** Corporate HQ migrations in 2006–2010 and corresponding revenues during the year of migration by corporate HQ city.

A larger number of companies migrated their HQs in the period 2011–2018. The destinations of these migrations also changed to some extent. In this time period all the continents except for Latin America were involved. The largest number of companies (7) moved to London—mostly from America and Europe. Their total revenues equaled 89.40 billions USD. Dublin was also a popular destination, as it had been in the previous study period. Six companies moved to Dublin, with a total revenue of 81.49 billions USD. Five of them originated in the United States—New York, Los Angeles, Cleveland, Minneapolis, Allerton. It is noteworthy that traditional tax havens were not successful during the studied period of time. Only one company moved from Dublin to Hamilton. In the years 2011–2018 not only Irish and Swiss cities were popular migration destinations, but also others such as Amsterdam, Luxembourg, Bilbao, and Monaco. These cities were able to attract large firms from abroad; however, these were mostly firms that had originally been headquartered in the European Union. Few foreign companies moved to cities outside of Europe. Examples included Sydney, Johannesburg, and Hamilton. Some companies simply moved from one city to another within the same country. Examples include moves from Baton Rouge to Charlotte, Springfield to Hartford, and Boston to Marlborough, all in the United States (Figure 6).

The main relocation destination for American companies has been Dublin, Ireland. This is due to economic considerations and cultural issues [51]. Many American CEOs are of Irish extraction and tend to factor in traditions associated with Ireland when examining relocation options [52]. Another major direction for American companies is London and Dublin, for whom it represents the gateway to Europe. This trend is also associated with the close relationship between the USA and UK, although Brexit may change this pattern.

The situation was different in Chinese cities. No major company had moved from abroad to China. Only one company moved from Hong Kong to mainland China. No Chinese company moved abroad either. Four companies moved within China—two migrated to Beijing and one to Binzhou and Chongqing each.

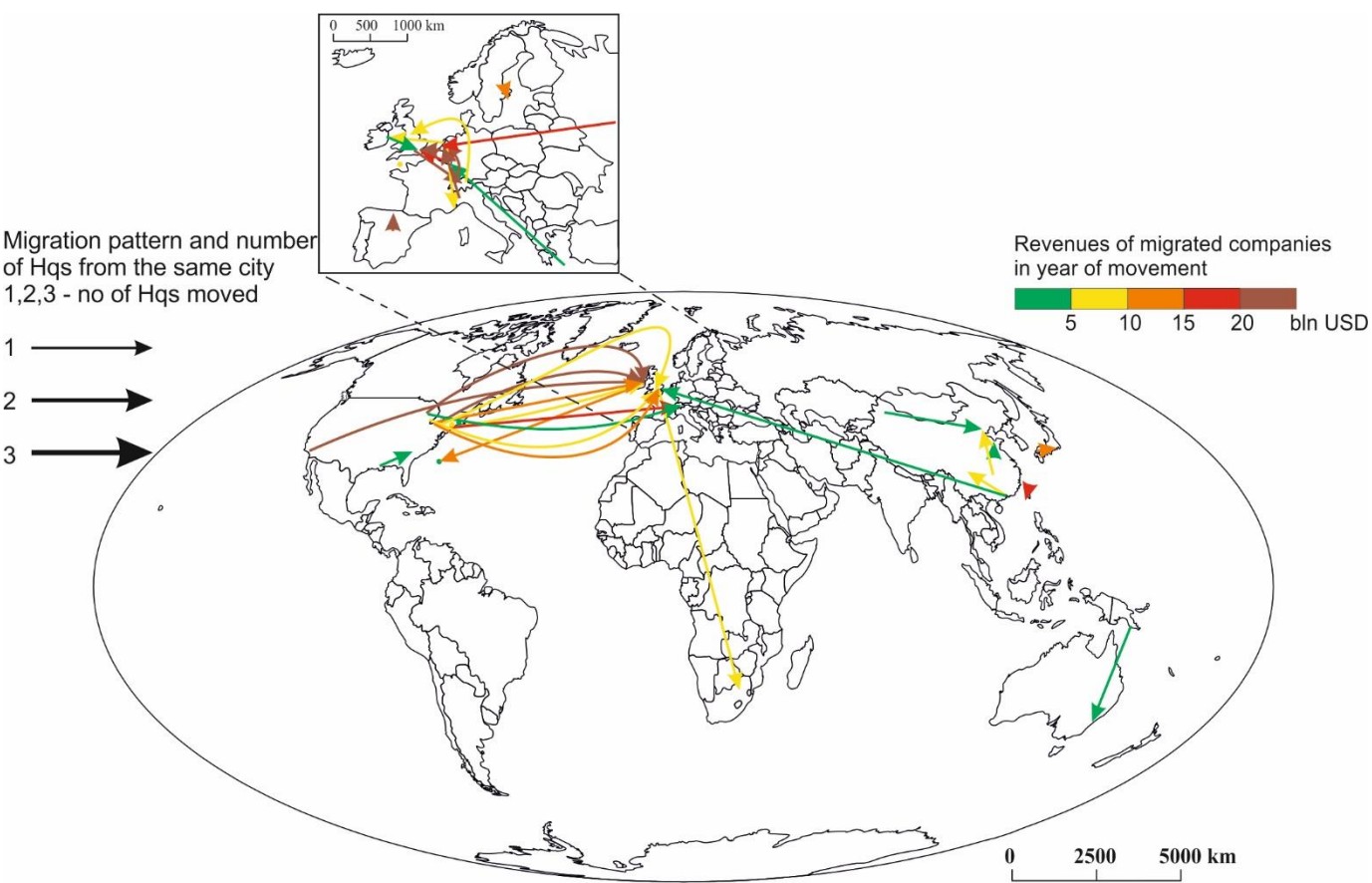

**Figure 6.** Corporate HQ migrations in 2011–2018 and corresponding revenues during the year of migration by corporate HQ city.

In the years 2011–2018 a total of 26 cities became destinations of HQ migration, while in the previous studied period their number was only five. This is an increase of 500%. In addition, previously dominant destination cities became less dominant. The two most attractive cities in the period 2011–2018 (London and Dublin) captured 32.53% of migrating companies with a 30.05% share in revenue. The corresponding values for the previous studied time period were 61.53% for Dublin and 71.17% for Zurich. These two cities were very attractive in the period 2006–2010, drawing companies above the revenue average for all migrating firms, leading to a higher share in total revenue for the two cities. On the other hand, the second study period was characterized by a decline in the attractiveness of London and Dublin as well as a decline in revenues per company. This led to a lower share for these two cities in total corporate revenues calculated for migrating firms (30.05% vs. 32.5%).

## 5. Discussion

The second decade of the 21st century has seen changes in the distribution of the command-and-control function at the city level and country level. Even though companies headquartered in Tokyo produced the largest revenues among the hundreds of cities listed by Forbes, the city is being rapidly outpaced by Beijing and its group of powerful corporations. Previous leaders such as New York, Paris, and London now occupy downgraded spots on the list of top world cities. The global crisis of 2008 had most strongly impacted companies in New York and London. The two cities are intimately linked and form the largest intercity network in the world, sometimes called NY-LON [53]. This strong linkage in itself may have triggered a stronger downward spiral during the financial crisis of 2008 than would have occurred in less globalized cities. Beijing began to outpace other major cities in this area in 2006 and has ranked as the strongest city in this respect since 2015.

Beijing outpaced former leaders including New York, Tokyo, London, and Paris mainly thanks to its financial, energy, and construction sectors [37].

The period after the global financial crisis of 2008 was better utilized by American companies that began to experience increases in revenue already in 2010, while companies in the European Union continued to see losses until 2018. The highest rates of growth were recorded in China, where growth of 1000% was not unusual. A gap was also observed in terms of the command-and-control function in EU and US cities. The C&C function of EU cities continued to decline in the years after the 2008 global financial crisis, while that of American cities continued to grow, leading to an increasingly large chasm between the two economic regions. Yet the largest gains were produced by Chinese firms and Chinese cities, which did not suffer excessively because of the crisis and managed to realize a more than a tenfold gain in the 13 years since the crisis. According to research by Li and Feng (2020), the large gains made by Chinese cities are not only due to their market-based success, but also due to political and economic support provided by the national government. This synergy has allowed cities such as Shanghai, Beijing, Shenzhen, Nanjing, and Suzhou to transform into major investment cities. Thus Chinese corporations cannot simply decide to change the location of their corporate headquarters without approval from China's national government. Ultimately, the growing magnitude of the command-and-control function of Chinese cities is due to the improving financial performance of companies located in these cities [37].

Our research has shown that the relocation of the headquarters of companies that generate the command-and-control function of cities does not produce a meaningful effect on revenues in these cities and on their C&C function. Only 53 companies relocated their headquarters to another city or country during the study period and their share of the total revenue of the 2000 Forbes companies studied herein equaled only 0.16%. However, it is noteworthy that the number of such companies and their revenues did increase during the years 2016–2018. While the first years of the study period were characterized mostly by relocations from the US and tax havens to Switzerland and the EU, in later years the number of countries of origin and destination countries increased. High-tech companies are more likely to relocate their corporate headquarters for non-economic reasons including their use of a global production network [46]. Biotech companies migrate frequently to Ireland and Switzerland due to lower taxes and availability of highly qualified workers [45].

Another readily observable process is the exit from tax havens due to the creation of severe restrictions on companies that take advantage of such locations. This process began after the global financial crisis of 2008 and has continued ever since in the European Union and elsewhere in the First World [50]. The main relocation target in the EU is Dublin, Ireland, where many American firms tend to migrate. This is due to economic reasons, but also sentimental reasons associated with the ethnic origin of many Irish-American CEOs who value Irish culture [52]. Another major relocation target is London, which for many Americans serves as the gateway to Europe. This is the case because of the close relationship between the United States and United Kingdom, although the problem of Brexit may complicate this relationship in the future. Yet another important region is the country of China whose rapid economic growth is fueling huge gains in revenue and increases in the strength of the command-and-control function, even though the country remains perceived as a source of cheap labor. Companies from around the world have been investing in China for decades and this investment further increased after the global financial crisis of 2008 [54]. However, our research has shown that the largest global corporations are not moving their corporate headquarters to China. Companies that didn't move their headquarters to China. Only 5 companies have moved their headquarters within China (included one from Hong Kong).

In the case of multinational companies (also in the case of many domestic companies), high-level decisionmaking and production are frequently geographically separated. As an effect the 1970s oil crisis, U.S., Western European, and Japanese companies have tended to relocate manufacturing to developing countries such as China, Indonesia, and Mexico

(even CEE countries) (see the term NIDL). However, high-level decisionmaking (also the command-and-control, or headquarters function) has remained in developed countries, usually at the original site of the company (where previously most of the production had been located as well) or in the CBD of a global city. Our findings demonstrate that if a company decides to relocate its corporate headquarters, it either relocates it from city A to city B within the same country (e.g., in the U.S., from one state to another), or from city A to city B which is located in a tax haven (i.e., Ireland, Bermuda, and selected cantons of Switzerland). There is not a single example of a major company where high-level decision making had geographically followed the production function. In this paper, we focus on high-level decisionmaking and map cities from which the global economy is commanded.

## 6. Conclusions

The 21st century has witnessed immense growth in the Asian markets, which includes growth in the strength of the command-and-control function of cities on this continent as well as increases in the number of their international linkages. The largest companies from China are slowly outpacing a host of their counterparts from wealthier regions such as the USA, Japan, and the European Union in terms of financial performance. If this trend continues to hold, the China will outpace the USA in terms of the command-and-control function soon. The USA has been the leader in this area for the last dozen years or so. Existing studies have focused on the spatial distribution of the command-and-control function and rates of change therein based on the number of corporate HQs present in a given city, regardless of whether these corporations had been there for years or are new to the city.

Our research has shown that the strength of the C&C function in each city is generated by companies whose headquarters have been in the city for quite some time. The number of companies that did relocate their headquarters was rather small. The conclusion is that cities are not good at attracting corporate headquarters via marketing efforts. Many cities can attract branches and plants, but not headquarters. The more typical pattern may be observed in China, where rapid growth and increases in the strength of the command-and-control function are facilitated by a huge domestic market, government support, and revenue growth at companies that have been in the Chinese market for quite some time already.

In recent years some companies have moved their headquarters and their revenues have in fact risen, which may motivate at least some cities to argue for corporate HQ relocation. This may help large companies from given city expand its economy and also increase its citys level of prestige. More broadly may cause change the spatial distribution of the command and control function across the world. Thus, it is important to help track the location of corporate headquarters, and not merely assess company financial performance. The results of the present study and the conclusions provided herein open the door to a new research question: Which key factors determine the location of the command and control function of cities? This represents the starting point for a deeper analysis of the issues already discussed in the present paper.

**Author Contributions:** Conceptualization, P.R. and G.C.; Data curation, A.W.-R.; Formal analysis, S.D. and A.W.-R.; Investigation, P.R.; Methodology, S.D.; Resources, S.D.; Software, S.D.; Validation, A.W.-R.; Visualization, S.D.; Writing—original draft, P.R.; Writing—review & editing, P.R. and G.C. All authors have read and agreed to the published version of the manuscript.

**Funding:** This research received no external funding.

**Institutional Review Board Statement:** Not applicable.

**Informed Consent Statement:** Not applicable.

**Data Availability Statement:** https://www.forbes.com/lists/global2000/#a828b515ac04 (accessed on 14 August 2020).

**Conflicts of Interest:** The authors declare no conflict of interest.

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
