# Peer review of "Exploring the Shifting Geographical Pattern of the Global Command-and-Control Function of Cities"

_sustainability, doi:10.3390/su132212798_

Round 1
Reviewer 1 Report
Komentarze do autorów:
1 STRESZCZENIE/Wprowadzenie
Linie: 8-11 i 147-151 Cel we wstępie (linie 8-11) różni się od linii 147-151. Należy wyjaśnić
2. Dane i metody
Dlaczego we wzorze matematycznym używa się liczby cztery, a nie innej?
Należy dodać opis wartości CCI. To sprawi, że ta część gazety będzie bardziej przejrzysta
- Zmiany...
Lines: 188-190 Figure 1 should be named Table 1. CCI values of the cities should be added to show the differences between cities. It is a normal ranking of cities, and it is not known based on which criterion cities are compared (what indicator size).
Line: 191 „The number five spot was taken by Beijing in 2007, which remained there until 2011”.
The sentence is not clear. Is it about indicator value or dynamics of the phenomenon?
Lines: 198-199 Numbers of the figures should be checked (Lines 220, 241, 257, 307, 324)
- Changes in corporate ....
Line: 256-257 The chapter should not start from the figure
Line 265: it should be Table 1
Lines 277-278: This sentence is debatable. What does it mean? The article does not examine the affiliation of the corporation from which country they are. Can they not have mixed capital: be transnational?
Lines 278-279 are not clear. If corporations are relocating within the same country, does that make a global difference? How should this be understood? If so, in what sense?
Lines 302-304: Does the large revenues growth of Chinese cities go from headquarters change or their good financial performance? Do the hqs change doesn’t matter for them?
Lines: 326 – 331: It is confusing. In the previous text, Dublin was mentioned
Lines: 306 and 324 The arrows on figures 6 and 7 should be clarified
- Discussion
Lines 445-446: it should be clarified and explained
„This pattern also holds in other countries, and it may be argued that capital has nationality”.
Lines 448-449: What cities are considered in this sentence? This conclusion is debatable
„Biorąc pod uwagę tę rzeczywistość, jest mało prawdopodobne, aby miasta były w stanie przyciągnąć wiele siedzib korporacji i być może powinny bardziej skoncentrować się na przyciąganiu oddziałów i zakładów produkcyjnych”.
- Wnioski
Wyniki i wnioski otwierają nowe naukowe pytanie: jakie czynniki determinują lokalizację funkcji dowodzenia i kontroli w poszczególnych miastach. Takie zdanie mogłoby zostać dodane w rozdziale podsumowującym.
Author Response
Dear Reviever,
We thank you for your useful comments and we hope that we have managed to improve the quality of the paper, and now it meets your expectations.
Best regards
Authors

Reviewer 2 Report
This paper proposes to examine the drivers of growing command and control function of cities by introducing a composite indicator composed of the following variables: revenues, profits, market value, and assets. The paper considers companies selected from Forbes Global 2000.
The main findings show that traditional centers of corporate headquarters (New York, London, Tokyo) has been lessening for a while, whereas cities located in developing countries and China (Beijing in particular) in the first place have been occupying an increasing position in the global command and control. Migration of headquarters does not impact the change of cities’ command and control functions.
The paper’s biggest advantage is to be very clearly written and with all parts very well explained. Nevertheless, it could benefit from a few corrections. The remarks/critics are listed bellows:
- The literature review could include more citations of C. Rozenblat working in the field of firm networks and from Zdanowska 2020 (Environment and Planning B paper) regarding CEE ;
- “Today, there exists” – please find a better formulation;
- In the abstract, the word ‘function’ is missing: “whether the growing command and control of a particular city”. Without the word function, growing loses sense;
- What are the negative points of the Forbes Global 2000 data? What is missing? What are the advantages comparing to other datasets?
- Which years does Forbes 2000 include? How did you compare 2006 and 2018? All this needs to be better explained in the data section. Is Forbes complete in the same way every year? How is it collected?
- How can we criticize the category “Western Europe” today? It is EU?
- How can you define “growth in command and control”? This is not clear in some part of the text and how are you justifying the growth in time (as you need a point of comparison, is it 2006?);
- “The data was then assigned to cities with corporate headquarters” – this must be explained further:
- How are metropolitan areas defined and how was the aggregation done for worldwide metropolitan areas? Which dataset was used for that? From my knowledge, harmonized data on metropolitan areas at the worldwide scale is very scarce.
- Definition of the index: the authors need to clearly define what do they understand by profits (difference with revenues), assets and market value et justify the difference and the contribution of such variables (and not other ones as benefits or turnover or number of controlled firms/links) to measure the command and control function.
- Generally speaking the Data and Methods section should be enhanced;
- How Figure 6 is taking into account all the period 2006-2010? Please explain the measurement;
- Migration of headquarters vs migration of centres of production? Make a critic about analysing only HQ in discussion
- In general this is a very interesting contribution to literature
Author Response
Dear Reviever
We thank you for your useful comments and we hope that we have managed to improve the quality of the paper, and now it meets your expectations.
Best Regards
Authors

Reviewer 3 Report
The authors proposed a formula to describe the CCI of a city. The formula is quite simple with all the same weight and the selected four indexes. However, the city is quite complex and it can not be described with such a simple index. More factors should be considered. Also, the authors compared their results based on five major cities in the world. However, these cities have completed different political and economic conditions. It is difficult to compare them just with the proposed index. Therefore the conclusions are not applicable. It is suggested that the authors can focus on the cities in the same country and discuss the effectiveness of the selected index for the CCI.
Author Response

(The authors gave the same response as above.)
